# *Paraburkholderia* Symbionts Display Variable Infection Patterns That Are Not Predictive of Amoeba Host Outcomes

**DOI:** 10.3390/genes11060674

**Published:** 2020-06-20

**Authors:** Jacob W. Miller, Colleen R. Bocke, Andrew R. Tresslar, Emily M. Schniepp, Susanne DiSalvo

**Affiliations:** Department of Biological Sciences, Southern Illinois University Edwardsville, 44 Circle Drive, Edwardsville, IL 62026, USA; jm6664953@gmail.com (J.W.M.); colwagn@siue.edu (C.R.B.); antress@siue.edu (A.R.T.); eschnie@siue.edu (E.M.S.)

**Keywords:** social amoeba, *Dictyostelium discoideum*, symbiosis, endosymbiont, infection

## Abstract

Symbiotic interactions exist within a parasitism to mutualism continuum that is influenced, among others, by genes and context. Dynamics of intracellular invasion, replication, and prevalence may underscore both host survivability and symbiont stability. More infectious symbionts might exert higher corresponding costs to hosts, which could ultimately disadvantage both partners. Here, we quantify infection patterns of diverse *Paraburkholderia* symbiont genotypes in their amoeba host *Dictyostelium discoideum* and probe the relationship between these patterns and host outcomes. We exposed *D. discoideum* to thirteen strains of *Paraburkholderia* each belonging to one of the three symbiont species found to naturally infect *D. discoideum*: *Paraburkholderia agricolaris, Paraburkholderia hayleyella,* and *Paraburkholderia bonniea*. We quantified the infection prevalence and intracellular density of fluorescently labeled symbionts along with the final host population size using flow cytometry and confocal microscopy. We find that infection phenotypes vary across symbiont strains. Symbionts belonging to the same species generally display similar infection patterns but are interestingly distinct when it comes to host outcomes. This results in final infection loads that do not strongly correlate to final host outcomes, suggesting other genetic factors that are not a direct cause or consequence of symbiont abundance impact host fitness.

## 1. Introduction

Amoebas are major predators of bacteria, engulfing them via phagocytosis and digesting them via phagosome-lysosome fusion [1,2,3,4]. This serves as a powerful selective pressure for the evolution of anti-predation defense mechanisms by bacteria. Some bacteria produce secreted toxins to repel or kill amoeba predators [5]. Alternatively, bacteria that can enter through phagocytosis but evade subsequent digestion gain access to an attractive intracellular niche and protective transmission strategy [6,7,8,9]. *Dictyostelium discoideum* is a social amoeba that serves as a natural host to a variety of bacterial endosymbionts and a model host for important human pathogens [10,11]. As such, *D. discoideum* has been particularly useful for studying the mechanistic underpinnings of bacterial infection processes and pathogenesis [3,12,13,14,15].

*D. discoideum* is a microbial eukaryote with single-cell and multicellular life stages. In the vegetative stage, single-celled amoeba prey on bacteria and divide by binary fission. When amoeba cells are populous and food bacteria have been depleted, amoebas co-aggregate to form a multicellular migratory slug. Slugs seek a suitable location and then differentiate into a fruiting body consisting of a globular sorus held above an erect stalk. The stalk consists of sacrificial cells and the sorus contains hardy spore cells which once dispersed to new locations will germinate into vegetative amoeba to continue the life cycle [16].

*D. discoideum* has effective innate immune mechanisms that function to resist or eliminate pathogens during vegetative and multicellular stages [4,17,18,19,20,21,22]. Despite the many challenges a bacterium faces to effectively colonize *D. discoideum*, bacteria belonging to the *Paraburkholderia* genus are prevalent and stable endosymbionts of *D. discoideum* [23,24,25,26,27,28,29]. These bacteria comprise three novel species, *P. bonniea, P. agricolaris,* and *P. hayleyella,* which intracellularly infect *D. discoideum* amoeba and persist through the social stage to continue transmission in spore cells and sori contents [25,26,29]. These species and related members have previously been described as belonging to the *Burkholderia* genus but have since been proposed as species of the recently nominated *Paraburkholderia* genera [30].

*Paraburkholderia* infections incur benefits and detriments to their hosts depending on host-symbiont genotypes and environmental context [23,24,26,27,28,31,32]. Under typical lab culture conditions, some *Paraburkholderia* strains decrease spore productivity while others appear to have little or no impact on the host population. In general, infection with *P. hayleyella* appears to be the most detrimental to host spore productivity, followed by *P. agricolaris,* while infection with *P. bonniea* does not appear to significantly reduce spore productivity [25,26,28,29]. However, only a small subset of strains from each species have been extensively investigated. In addition to determining how consistent strains are within species on host fitness, we wished to quantify and compare final intracellular infection patterns across strains. Ultimately, we hypothesized that infection patterns would correlate to host outcomes, with strains that infect higher numbers of hosts and/or at higher densities incurring the highest fitness costs to their host populations.

To explore the diversity of infection patterns across symbiont strains, we quantified three end-point infection outcomes from a collection of thirteen *Paraburkholderia* symbionts. These include infection prevalence as the percent of intracellularly infected spores in the population, infection density as the number of intracellular symbiont cells per infected spore, and total host spore productivity as a measure of host fitness following infection. We also infected hosts using two different *Paraburkholderia* exposure doses for each strain. This allowed us to examine how exposure dose impacts final outcomes as well as to expand potential variability in infection outcomes to probe the relationship between infection patterns and host fitness. Our results demonstrate that there is significant variability in infection patterns and outcomes across strains. There is a very weak trend indicating that higher infection phenotypes relate to worse outcomes, however, this is not significant. Strains within the *P. hayleyella* species have similarly high infection phenotypes yet show wide variability in their impact on host fitness. Although we did not find evidence for a secreted toxin in mediating these differences, this variation suggests that other virulence factors, not implicitly associated with infection quantity, differ between these strains.

## 2. Materials and Methods

### 2.1. Bacteria and Amoeba Culture Conditions

We cultured *Paraburkholderia* spp. and *Klebsiella pneumoniae* on SM/5 agar medium [(premix from Foremedium) 2 g peptone, 0.2 g yeast extract, 2 g glucose, 1.9 g KH_2_PO_4_, 1.3 g K_2_HPO_4_ 3H_2_O, 0.49 g MgO_4_ anhydrous, and 17 g agar per liter] at room temperature. All *Paraburkholderia* strains used along with the location isolated are listed in Appendix A. We used RFP labeled *Paraburkholderia* strains and a GFP labeled strain of *K. pneumoniae* for all experiments, generated as previously described [26,28]. To prepare bacterial cultures for amoeba co-culturing, we suspended bacteria in the stationary phase from SM/5 agar medium in KK2 buffer (2.25 g KH_2_PO4 (Sigma-Aldrich, St. Louis, MO, USA) and 0.67 g K_2_HPO4 (Fisher Scientific, Waltham, MA, USA) per liter) to an OD_600nm_ of 1.5. For uninfected conditions, we plated 10^5^
*D. discoideum* spores with 200 µL of *K. pneumoniae* suspension set at an OD_600nm_ of 1.5. For *Paraburkholderia* exposure conditions, we mixed each *Paraburkholderia* strain suspension with *K. pneumoniae* suspension (both set at an OD_600nm_ of 1.5) at a 5% and 0.5% *Paraburkholderia/K. pneumoniae* (vol/vol) ratio as indicated. We then plated 10^5^
*D. discoideum* spores with 200 µL of these bacterial mixtures on SM/5. Based on colony-forming quantification, we estimate an average of 6 × 10^7^ symbiont cells added to plates for the 5% exposure condition and 6 × 10^6^ symbiont cells added for the 0.5% exposure conditions. We incubated amoeba cultures at room temperature under LED strip lights for fruiting body formation. Samples were harvested from these plates after five days of incubation (for confocal microscopy) or seven days of incubation (for infection prevalence and spore count assays using the flow cytometer). We used the *D. discoideum* strain QS18 for all experiments, which is a natural isolate collected by the Strassmann–Queller lab.

### 2.2. Confocal Microscopy and Infection Density Assay

To quantify the number of *Paraburkholderia* cells within spores, we imaged spores via confocal microscopy. We collected developed sori into 100 μL of KK2 buffer with 1% calcofluor-white to stain the cellulose spore wall, vortexed well, placed the spore suspension on #1.5 cover glass-bottom culture dishes (70674-52 from Electron Microscopy Sciences, Hatfield, PA, USA) and flatted them with a 2% agarose overlay. We imaged spores with an Olympus Fluoview 1000, (Olympus Life-Science, Center Valley, PA, USA) confocal microscope using the DAPI and Cy3 channels to image calcofluor and *Paraburkholderia*-RFP respectively. Z-sections were taken every 0.5 microns with an average of 2 at 1024 × 1024 resolution and images were processed using FIJI (Bathesda, MA, USA) (imagej.net/Fiji). We quantified infection density by counting the number of *Paraburkholderia* cells for 20 spores per replicate, using at least two independent image replicates for each treatment. We measured cell size and the intracellular area occupied by each *P. hayleyella* strain using the scale measurement and particle analyzer function in FIJI from z-projections of confocal micrographs for 10 cells and 10 spores, respectively.

### 2.3. Total Spore Productivity

To measure total spores, we harvested entire plate contents into KK2 buffer with 0.1% NP40 and vortexed. We diluted samples 10 or 20-fold for analysis with the BD Accuri C6 Plus flow cytometer (BD Biosciences, San Jose, CA, USA). We established spore gating using an XY scatterplot of FSC-H and SSC-H and used the number of recorded particles to calculate total spores for the total plate suspension.

### 2.4. Infection Prevalence

To quantify the percent of spores infected by *Paraburkholderia*-RFP in each condition, we harvested approximately 5 sori per plate into 500 µL of KK2 buffer and analyzed 100 µL through the flow cytometer. After gating spores, we determined the percent of RFP positive spores using a histogram of PE-A intensity by particle count. Uninfected controls were analyzed to establish an accurate vertical gating line between fluorescent and non-fluorescent boundaries.

### 2.5. Infection Stability

To determine the stability and prevalence of *Paraburkholderia* infection in spore populations over multiple social cycles, we plated spores as previously described with *Paraburkholderia* strains (Bh11, Ba70, Bb859) at the indicated exposure doses (5% and 0.5% by volume with *K. pneumoniae*). Three replicates were analyzed in parallel and each replicate was plated in duplicate. Once the initial infection was established, we incubated plates for five days before sampling and transferring to new plates. We performed transfers by plating 10^5^ spores from the previously developed plate with 200 µL of a 1.5 OD_600nm_ suspension of *K. pneumoniae* on SM/5 agar medium. We determined population infectivity by analyzing spores developed from each transfer using flow cytometry as previously described.

### 2.6. Supernatant Toxicity

To determine whether filterable metabolic products secreted by *Paraburkholderia* were detrimental to the fitness of *D. discoideum*, we cultured each *Paraburkholderia* strain overnight in 3 mL SM/5 broth. After 24 h of growth, we centrifuged cultures at 13,000 rpm for two minutes and filtered supernatants through 0.22 μm filters. We then plated 10^5^
*D. discoideum* spores with 200 µL *K. pneumoniae* at an OD_600nm_ of 1.5 and 500 μL of the *Paraburkholderia* culture medium filtrates. We analyzed total spore productivity after a week of incubation using flow cytometry as previously described. We tested four or more replicates for each bacterial strain and used *K. pneumoniae* filtrates as a control.

### 2.7. Phylogenetic Tree Construction

We adapted a phylogenetic tree from Haselkorn et al., 2019 using concatenated multi-locus sequencing data from dryad (https://datadryad.org/stash/dataset/doi:10.5061/dryad.g23s038). We selected the thirteen *Paraburkholderia* symbiont strains used in this study along with a collection of related environmental species and two pathogenic *Burkholderia* species for rooting. We imported sequences into Mega7, aligned with MUSCLE, and constructed a neighbor end-joining tree with 1000 bootstrap replicates.

### 2.8. Statistical Analysis

All statistical analysis was performed in R 3.6.0. Comparisons were done using two-way ANOVA’s with symbiont strain and exposure dose as independent variables for infection pattern experiments. Comparisons of *P. hayleyella* cell size, intracellular area occupied, and supernatant toxicity were analyzed using one-way ANOVA’s. When appropriate, multi-comparisons were made among strains using a post-hoc Tukey test. To measure the correlation between infection patterns and total spore productivity, we used a Kendall rank correlation test. We estimated the total infection titer by multiplying the infection prevalence mean by the infection density mean. To estimate population-wide total symbiont abundance we multiplied each spore productivity replicate by an infection titer metric (as the multiple of the mean of percent infection prevalence by the mean of infection density). We used a Kruskal–Wallis test to compare the total estimated endosymbiont abundance across strains.

## 3. Results

### 3.1. Different Paraburkholderia Symbionts and Exposure Doses Lead to a Range of Final Infection Patterns

To address the association between intracellular symbiont loads and their corresponding fitness impact on host populations, we quantified infection patterns for thirteen *Paraburkholderia* strains in a single host isolate. We hypothesized that if infection outcomes were variable across strains, we would be able to interrogate the relationship between infection patterns and host outcomes. These thirteen *Paraburkholderia* strains represent the three-known natural *Paraburkholderia* symbiont species of *D. discoideum* [29]. Phenotypic, metabolic, and phylogenetic comparisons have been done for representative strains of each of these species [29]. For all thirteen strains used in this study 16S rRNA and *phaC*, *atpD*, *gltB*, *lepA*, and *trpB* house-keeping gene sequences have been previously analyzed for the construction of a phylogeny [25]. Here, we adapted this phylogenetic tree to represent the relationship between the strains selected for this study along with related *Burkholderia* species (Figure 1). No within-species nucleotide variation was detected for the *P. hayleyella* strains. For *P. bonniea*, strain Bb859 was distinct from Bb433 and Bb395 strains. For *P. agricolaris*, Ba70 and Ba161 had identical gene sequences to one another but varied from Ba31 and Ba159 (which were identical to each another). Although sequence typing did not reveal significant variability across all symbiont genotypes within species, sequences likely vary at other loci that may be important for imparting unique infection phenotypes.

To visualize symbiont infections, all strains were chromosomally labeled with RFP. To establish infections, we exposed 10^5^
*D. discoideum* spores to each *Paraburkholderia-*RFP strain at a 5% or a 0.5% concentration (mixed with 95% or 99.5% *K. pneumoniae* food bacteria, respectively). After fruiting body formation (five to seven days post-plating), we measured intracellular infection patterns and host spore productivity.

To quantify the infection prevalence of intracellular *Paraburkholderia*, we quantified the percent of spores containing fluorescent bacteria via flow cytometry (Figure 2). We found that the prevalence of intracellular *Paraburkholderia* was significantly different according to strain identity (df = 12, F = 28.865, *p* < 0.001) and exposure dose (df = 1, F = 24.329, *p* < 0.001). There is also a significant interaction between strain and dose (df = 12, F = 6.845, *p* < 0.001). Infection prevalence was also significantly associated with *Paraburkholderia* species identity (df = 2, R = 47.86, *p* < 0.001) with *P. agricolaris* infecting significantly less total spores (average of 34.4% RFP positive spores) than *P. hayleyella* or *P. bonniea* (with an average of 81% and 74% RFP positive spores, respectively). For the 0.5% exposure conditions, we observed large variation in results among replicates. The reason for this is unclear, perhaps when starting with smaller concentrations of symbiont cells, slight variation in initial plating volumes lead to a large variation in final symbiont numbers as they amplify (and potentially compete with food bacteria). Interestingly, the pattern of changes in symbiont prevalence observed between 5% and 0.5% exposure doses is not consistent across strains. We expected that prevalence would be lower at the 0.5% exposure dose as there would be fewer symbiont cells to initiate infections. We observe this trend for Bh11, Bh171, Bh22, Bh530, Ba70, Bb395, and Bb433. However, for strains Bh155, Bh21, Ba161, and Bb859 there is no change in prevalence between dosage conditions. This could reflect that the number of infective units is already at the threshold level for both exposure conditions. What is most puzzling is the unexpected increase in symbiont prevalence for Ba159 and Ba31 at the lower exposure dose. One possible explanation for this could be that these symbionts induce host cell lysis at higher volumes resulting in a decrease in final infected hosts following development.

To determine the final intracellular density of *Paraburkholderia*, we counted the number of fluorescent bacteria within individual spores using confocal microscopy (Figure 3). We found that *Paraburkholderia* strain identity and exposure dose is associated with significant differences in endosymbiont density (df = 12, F = 84.578, *p* < 0.001 and d = f1, F = 11.838, *p* < 0.001, respectively). There is also a significant interaction between strain and dose on symbiont densities (df = 12, F = 4.167, *p* < 0.001). Intracellular infection density was also significantly associated with *Paraburkholderia* species identity (df = 2, F = 309, *p* < 0.001). Cells of *P. agricolaris* strains accumulated the most within individual spores (average of 12.32 bacteria per spore), with *P. hayleyella* strains accumulating to moderate intracellular densities (average of 7.46 bacteria per spore), and *P. bonniea* strains accumulating the least (average of 3.45 bacteria per spore).

We next quantified final host fitness outcomes for each *Paraburkholderia* strain by quantifying total spore numbers after symbiont exposure (Figure 4). We found that *Paraburkholderia* strain and exposure dose is also associated with significant differences in final spore productivity (df = 13, F = 15.17, *p* < 0.001 and df = 1, F = 8.325, *p* < 0.005, respectively) and that there was a significant interaction between strain and dose on spore numbers (df = 12, F = 4.852, *p* < 0.005). Symbiont species identity was also significantly associated with total spore productivity (df = 3, F = 9.761, *p* < 0.001). *D. discoideum* infected with *P. bonniea* did not produce significantly different numbers of spores than uninfected *D. discoideum* (*p* = 0.31). However, the number of total spores produced after exposure to *P. hayleyella* or *P. agricolaris* species were significantly different from the number of spores produced by uninfected *D. discoideum* (*p* < 0.001) but did not significantly differ from one another species (*p* = 0.83). Within species, strains of *P. bonniea* were similar in their impact on host fitness, resulting in no significant difference in host spore productivity (df = 2, F = 0.383, *p* = 0.686). *P. hayleyella* and *P. agricolaris* strains produced more variable host outcomes within species, resulting in significant differences in host spore productivity (df = 5, F = 15.89, *p* < 0.001 and df = 4, F = 5.241, *p* < 0.005 respectively). For *P. hayleyella*, this pattern is particularly striking as some strains appear highly detrimental to host fitness while others appear commensalistic.

### 3.2. Final Symbiont Infection Properties do not Strongly Correlate with Host Outcomes

As we found that final intracellular infection patterns and host outcomes were variable across symbiont genotypes and exposure conditions, we were able to probe the relationship between these factors. To investigate the correlation between host fitness outcomes according to symbiont infection traits, we plotted the average spore productivity of *D. discoideum* against the average infection prevalence (Figure 5a) and the average infection density (Figure 5b) for each strain and exposure condition. We found that there was no significant correlation between infection prevalence and total spore productivity (r_τ_ = −0.0285, *p* = 0.834). For instance, at the 5% exposure dose, all *P. hayleyella* strains are highly prevalent in the host spore population, but Bh11, Bh155, and Bh171 severely reduce spore productivity while strains Bh21, Bh22, and Bh530 do not. When comparing intracellular infection density and spore productivity we found a significant, albeit very weak, negative correlation between the two (r_τ_ = −0.2934, *p* = 0.0325). This seems reasonable, as individual amoeba may be more negatively impacted by higher intracellular accumulations of symbionts. If final infection density within spores reflects a higher accumulation of symbionts in vegetative amoeba, an excess of symbiont cells likely induces amoeba death, lysis, or failure to differentiate into spores. However, some of the *P. hayleyella* strains that most dramatically reduce host fitness (Bh11 at both exposure doses and Bh171 at 5% exposure doses) only achieve middling intracellular density within spores.

We speculated that symbiont density and symbiont prevalence within a population should act synergistically in impacting host fitness. For instance, symbionts that accumulate to high densities and are also highly prevalent in the population should be more detrimental than those that still accumulate to high densities but are not highly prevalent. Thus, we calculated a “final infection titer” unit to combine these two infection patterns by multiplying the average intracellular density by the average infection prevalence for each symbiont exposure treatment. We then plotted the average total spore productivity by this infection titer unit to visualize the relationship between them (Figure 5c). Although we found that there was a weak negative correlation between infection titer and host spore productivity, this was not significant (r_τ_ = −0.225, *p* = 0.1043). Many strains that exhibited similar infection titer values had broadly different host outcomes. For instance, *P. hayleyella* strains Bh11 and Bh530 both have similar infection titer values but Bh11 dramatically reduces host spore productivity while Bh530 does not. These strains appear to only differ in their impact on spore productivity as they have similar infection prevalence and similar infection densities (Figure 5). Overall, this suggests that although there may be a weak relationship between infection patterns and host productivity, quantitative infection load is not the key player in mediating host outcomes.

Although intracellular symbiont titers do not strongly correlate to host outcomes, we did not consider the contribution of symbiont cell size to these factors. If different *P. hayleyella* strains vary in their cell size, this could explain their differential impact on host fitness despite reaching similar intracellular abundances. For instance, symbiont cells that are individually larger or that result collectively in occupying a larger intracellular area in host cells should exert higher fitness costs. To determine if these measurements relate to differential host impacts we measured the cell length (Figure 6a) and the total area occupied (Figure 6b) within host spores from confocal micrographs of *P. hayleyella* infected spores. We found significant differences in cell size and intracellular area occupied across *P. hayleyella* strains (df = 5, F = 3, *p* < 0.05 and df = 5, F = 12, *p* < 0.01, respectively). However, these differences did not unambiguously match with differences in host fitness. *P. hayleyella* strain B11 was significantly larger than strain B530, but all other comparisons between cell sizes were insignificant. For the intracellular area occupied, only strain B155 occupied a significantly larger area than the other strains, which were not significantly different from one another. B155 is indeed more detrimental to host fitness than most of the other *P. hayleyella* strains (except for B11) and the larger area occupied by this strain may exert a higher cost on host fitness. However, considering the spread of patterns across our collection of symbiont strains, general differences in intracellular abundance or area occupied within spores do not consistently connect to differences in host fitness.

### 3.3. Secreted Toxins not Detected from P. hayleyella Symbiont Supernatants

Since we found that the quantity of intracellular symbionts in *D. discoideum* populations is not significantly associated with host spore productivity, we investigated whether secreted toxins could explain differences in host outcomes. We specifically focused on *P. hayleyella* strains, as these resulted in the largest range in host spore productivity even though they generally reached similar intracellular infection quantities. For this assay, we harvested supernatants of overnight liquid cultures from each *P. hayleyella* strain and *K. pneumoniae* (as a control) set at equal optical densities. Culture supernatants were filtered through 0.22-micron filters to eliminate bacterial cells and then added to nutrient plates inoculated with *D. discoideum* spores and *K. pneumoniae* food bacteria. After seven days of incubation, we quantified total spore productivity (Figure 6). We found that all *D. discoideum* cultures produced similar numbers of total spores, with no significant differences between any of the supernatant exposure conditions (One-way ANOVA, df = 6, F = 1.661, *p* = 0.163). Thus, we were unable to detect the presence of secreted filterable toxins from *P. hayleyella* strains that could explain their differences in impacting host spore productivity during infection. Despite our inability to detect secreted toxins using the present assay, these results do not exclude the possibility that they may nonetheless play a role in the impact of *Paraburkholderia* strains on host fitness.

### 3.4. Symbiont Prevalence Plateaus over Multiple Social Cycles, Reaching Similar Levels Regardless of Exposure Dosage

We next wanted to address the stability of infection prevalence over multiple social cycles and how initial exposure dose impacts infection maintenance over time. To do this, we cultured *D. discoideum* spores with one strain representative for each *Paraburkholderia* symbiont species under 5% and 0.5% exposure conditions. After *D. discoideum* completed the first social cycle with exposure to *Paraburkholderia*, we serially transferred developed spores to new plates with *K. pneumoniae* alone for four more social cycles. At the culmination of each social cycle, we measured the prevalence of *Paraburkholderia*-RFP infections in the spore population (Figure 7). When *D. discoideum* are exposed to symbionts at a 5% dosage, infection prevalence only moderately changes over each transfer to stabilize at approximately 90%, 25%, and 80% infected spores for *P. hayleyella* Bh11, *P. agricolaris* Ba70, and *P. bonniea* Bb859, respectively. When *D. discoideum* are exposed to the *Paraburkholderia* strains at a 0.5% dosage, infection prevalence increases over the first two transfers and stabilizes to the same final prevalence levels as their 5% exposure dose counterparts.

### 3.5. Symbionts Capable of Achieving High Infection Loads with Minimal Host Damage Should Maximize Symbiont Population Abundance

Endosymbionts able to maximize infectivity while minimizing host damage should be the most successful from an evolutionary perspective. Using our data on symbiont prevalence, density, and total spore productivity we estimated total intracellular symbiont abundance for each strain as a proxy for their population-wide relative success. For this estimate, we multiplied each experimental replicate of total host spore productivity by the infection titer metric of each symbiont strain (as calculated by the average infection density and the average percent of prevalence). Considering the results of our serial transfer experiment and the consistency of replicates for exposure dose conditions, we focused on our data from the 5% exposure treatment to estimate total intracellular symbiont abundance (Figure 8). Estimates of population-wide intracellular symbiont abundance were significantly different across symbiont strains (chi-squared = 56.156, df = 12, *p* < 0.001). In general, *P. hayleyella* strains have the highest total abundances in the host population. Indeed, the most abundant symbionts appear to be those that are moderately prevalent and dense while exerting minimal costs on host spore productivity.

## 4. Discussion

We found significant differences in final infection phenotypes across *Paraburkholderia* symbiont genotypes. We detected a weak significant negative correlation between infection density and spore productivity and a very modest but insignificant negative correlation between total infections titers and spore productivity. Thus, our prediction that higher infection loads would convey higher costs to host populations was not strongly supported. Interestingly, for many of the strains, particularly among the *P. hayleyella* and *P. bonniea* isolates, we observe infection prevalence above 90% but with no measurable impact to host fitness. The intracellular infection density is relatively low for *P. bonniea* strains, which may make them less costly for hosts. However, many *P. hayleyella* strains (such as Bh21) have high infection densities despite little to no detrimental impacts on host fitness. Because these symbionts can achieve high infection quantities without compromising their hosts, they maximize their own total population size within host communities.

The ability to establish a high infection load without compromising host fitness in this system is puzzling as even in clearly mutualistic symbioses high symbiont loads are associated with contextual costs for their hosts. For instance, increased densities of defensive symbionts in aphid hosts correlates to strong declines in host fitness [33,34,35]. In plant-mycorrhizae symbiosis, symbiont abundance plays a dynamic role in host fitness for different traits [36,37,38]. However, hosts are not passive bystanders in this process and may have active mechanisms to modulate symbiont locations and densities according to their situational costs or benefits [39,40]. Mutualisms may initially emerge from different positions of power, with either the host or symbiont exerting more control over their partner in the beginning while checks and balances evolve over time [41,42,43]. The danger of symbiont overabundance is well illustrated with opportunistic pathogens, which are tolerable at low thresholds but promote disease following overgrowth [44]. In purely parasitic relationships, we would expect high symbiont load to be even less tolerable for hosts. For instance, total *Plasmodium* parasite biomass has been correlated to more severe malaria symptoms and host responses [45,46]. In our study, we specifically used culture conditions in which *Paraburkholderia* symbionts have no clear contextual benefit to hosts. *P. hayleyella* strains in particular have been shown to be most detrimental to host fitness and potentially the least contextually beneficial, although previous studies did not examine as many strains as we have here [26,27]. Even though symbionts seem unlikely to provide benefits in these assays, *D. discoideum* cells must be capable of tolerating a fairly high infection level provided symbionts lack other virulence traits. However, this still suggests that the infection loads we observe for these symbionts are below some critical threshold that would compromise hosts. Although we did not detect any detrimental impact of *P. hayleyella* culture supernatants on host spore productivity, the possibility that secreted toxins play a role in differential virulence remains plausible. Secreted toxins are costly to produce, and therefore may be regulated by chemical cues from nearby predators [47]. Thus, alternative bacterial culture conditions could enable the production of more active supernatants. In addition, a previous study observed a decrease in *D. discoideum* spore productivity after amoeba were developed on filters saturated with *P. hayleyella* supernatants [32]. This indicates that some symbiont strains do in fact secrete factors toxic to amoeba and that more sensitive assays are warranted to fully elucidate their contribution to host outcomes [48].

Several host and symbiont factors may be responsible for governing the final intracellular infection prevalence and density of *Paraburkholderia.* These may result from differential infection dynamics which include (but are not necessarily limited to): the number of infection events or infection units, intracellular replication rates, the efficiency of digestion, and lytic or non-lytic egress events [49]. Different rates of exposure do not seem likely to explain variable infection patterns as we estimate a high multiplicity of infection for all culture conditions. We plate 1 × 10^5^ host spores and an estimated average of 6 × 10^7^ and 6 × 10^6^ symbiont cells for the 5 and 0.5 percent exposure dose conditions respectively; equating to *Paraburkholderia* cells outnumbering *D. discoideum* 600 and 60 to 1. This, in addition to the fact that some of the *Paraburkholderia* strains infect almost the entire population of *D. discoideum* spores, suggest that the difference in symbiont prevalence is not due to a failure of *Paraburkholderia* to come in contact with amoeba cells.

*D. discoideum* may actively discriminate between bacteria resulting in differential contact and phagocytosis of specific symbiont strains [4,50,51]. Alternatively, *Paraburkholderia* strains may have distinct mechanisms to evade engulfment by amoeba cells, but this would not promote intracellular infection. Instead, *Paraburkholderia* symbionts have been shown to actively migrate towards *D. discoideum* [52]. Another possibility is that host amoeba can digest *Paraburkholderia* symbionts at different efficiencies. In general, *Paraburkholderia* symbiont strains cannot support *D. discoideum* development without additional food bacteria, but this does not entirely preclude low levels of symbiont digestion inadequate for developmental progression [25]. This possibility can be investigated in the future through the use of fluorescent microscopy and bacterial viability assays to measure intracellular killing rates of *Paraburkholderia* strains by vegetative amoeba [53]. If some symbionts only weakly inhibit digestion, it may be that this inhibition could be improved via the additive function of multiple symbiont cells entering amoeba simultaneously. This might also explain the observation that *P. agricolaris* strains exhibit the lowest prevalence rates in host populations but achieve the highest intracellular densities within infected spores.

Differential intracellular replication rates may help explain the somewhat paradoxical *P. agricolaris* infection patterns where infection prevalence is comparatively low, yet infection density is comparatively high. Perhaps replication fails to be regulated by either host or symbiont mechanisms once *P. agricolaris* establishes an intracellular infection. Indeed, during confocal imaging, we have occasionally witnessed a host spore swelling with an overabundance of intracellular *Paraburkholderia* to the point where the spore appears on the verge of lysis. If some infected cells lyse from over-replication of bacterial symbionts before they make it to the final spore stage, our end-stage analysis would only capture spores that were never infected (or that cleared their infection) and spores with high (but not yet overwhelming) levels of *Paraburkholderia*. Future experiments to quantify symbiont dynamics throughout host development will help to address these possibilities.

Infection densities may also be determined by the rate at which bacteria replicate once inside host cells, as regulated by either host or symbiont. Inhibition of intracellular replication is not uncommon, certain isolates of the *Burkholderia cepacia* complex survive within acidic vacuoles of *Acanthamoeba polyphaga* without replicating for long periods of time [54]. *Salmonella typhimurium* also significantly decreases its rate of replication inside host macrophages, reaching a quasi-dormant viable state that once outside macrophages resume normal replication [55]. Pathogens may also directly inhibit host cell apoptosis, which is an important process for clearing intracellular infections from multicellular organisms [56]. As *D. discoideum* is only transiently multicellular, it may not have not developed nor benefited from this form of innate immunity. Indeed, *D. discoideum* does not have the molecular machinery to undergo apoptotic cell death and cell-death instead occurs through autophagic processes [57,58,59]. These processes are conserved between *D. discoideum* and mammals, opening the possibility that autophagic cell-death mechanisms could be initiated by *Paraburkholderia* infections and differentially regulated by symbiont strains [60,61]. In line with this, some intracellular pathogens can be killed via autophagy (termed “xenophagy”) and their ability to manipulate this pathway plays an important role in their pathogenesis [62,63]. Thus, it will be fruitful to investigate the relationship between autophagy and *Paraburkholderia* symbiont infections.

One challenging observation to interpret was the pattern of strain-specific variation in final symbiont prevalence according to exposure dose. For some strains, prevalence decreases as one might predict at the lower exposure dose, while in others it remains consistent or even increases. The underlying mechanisms for this are unclear, but may ultimately be explained by genotypic differences that regulate infection dynamics. We also found that after multiple social cycles, intracellular prevalence plateaus to similar strain-specific levels despite initial exposure condition. Symbiont prevalence in 0.5% exposure conditions ultimately mirrors those of 5% exposure conditions. Thus, strain-specific infection thresholds may hold stable once achieved, with the 5% exposure condition immediately reaching this stable infection threshold. These results suggest that the 5% exposure condition is sufficient for quickly reaching a stable final infection level for experimental analysis. However, it would be interesting to see whether infection thresholds differ according to other contexts, such as alternative media or culturing conditions and in ecological settings. Ultimately, our results lay the groundwork for future investigations to clarify these dynamics and their genetic underpinnings.

Here, we uncovered variation across strains, particularly within the *P. hayleyella* species, that have not been previously appreciated. This is interesting because multi-locus sequence typing revealed no differences in the analyzed gene sequences within *P. hayleyella* strains [25]. However, this phenotypic variability suggests that there must be a difference in pathogenicity factors among these strains. Focusing additional experiments and genomic analyses on targeted *Paraburkholderia* strains would be useful in elucidating the mechanisms underlying differential virulence traits. In specific, genomic comparisons between Bh11, Bh155, Bh21, and Bh530, may be particularly fruitful in revealing why these strains vary dramatically in their impact on host fitness despite little variation in infection prevalence and density. Indeed, comparative genomics of related symbiont strains that diverge in host impacts have been fruitful in other symbiosis systems to illuminate phenotype-genotype associations, horizontal gene transfer events, and intracellular symbiont genome evolution [64,65,66]. Overall, the inherent genetic and phenotypic diversity of this system should provide compelling investigations into pathogenic versus commensal infection mechanisms, host responses, and their genetic basis.

## 5. Conclusions

We measured intracellular infection patterns for thirteen symbiont genotypes under two exposure doses in an amoeba host. This included the quantification of final infection prevalence’s in host populations (percent of infected spores), symbiont density within infected spores (number of symbiont cells per spore), and host fitness after symbiont exposure (total host spore productivity). We speculated that symbiont abundance would negatively correlate with host fitness, as higher symbiont loads would presumably be more costly for hosts. Although there was a weak correlation between host fitness and symbiont density, several symbiont strains could establish high infection abundances without compromising host spore productivity. This was particularly interesting as symbiont genotypes within the *P. hayleyella* species displayed similar infection patterns but produced very different impacts on host fitness. Previous multi-locus sequence typing of these *P. hayleyella* strains found no nucleotide variation, however our results show they can vary significantly in virulence. Our results also suggest that *D. discoideum* can tolerate a relatively high load of intracellular symbionts, and that symbionts capable of maximizing infectivity while minimizing virulence have a relative population advantage. These results will help inform future research into the underlying molecular mechanisms mediating symbiosis outcomes and on how the interplay between infection patterns and host outcomes impact evolutionary trajectories. 

## Figures and Tables

**Figure 1 genes-11-00674-f001:**
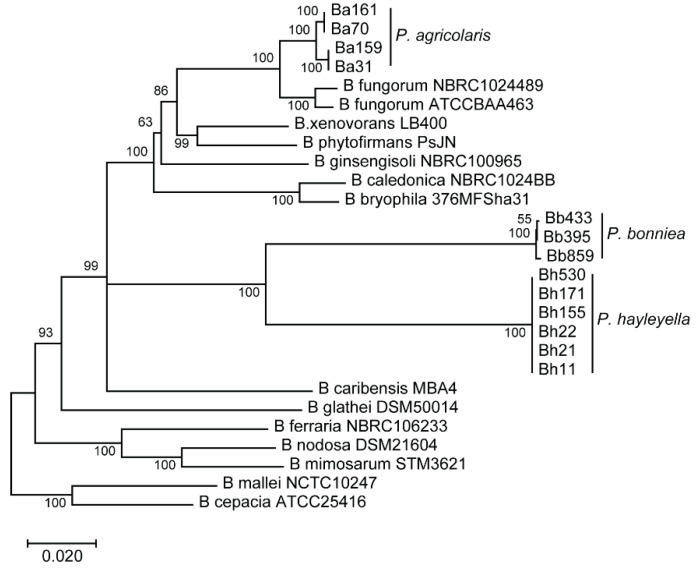
Phylogenetic tree of *Paraburkholderia* symbiont strains. Phylogeny of selected *Paraburkholderia* and *Burkholderia* strains based on multi-locus sequence typing of the 16s rRNA gene and the five housekeeping-genes *phaC*, *atpD*, *gltB*, *lepA*, and *trpB.* The tree was constructed using the neighbor-joining method in Mega7, with 1000 bootstrap replicates. Adapted from Haselkorn et al., 2019.

**Figure 2 genes-11-00674-f002:**
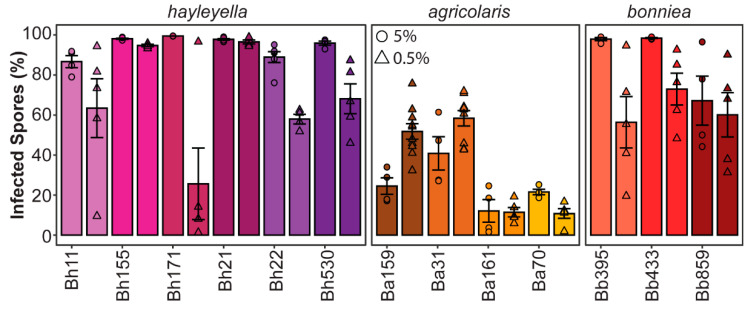
*Paraburkholderia* infection prevalence is variable across strains. Final percent of *Paraburkholderia*-RFP positive spores from fruiting bodies developed after exposure to 5% (circles) or 0.5% (triangles) of the indicated *Paraburkholderia* strain. Points represent individual replicate, bars indicate mean, and error bars as the standard error of the mean (*n* ≥ 4).

**Figure 3 genes-11-00674-f003:**
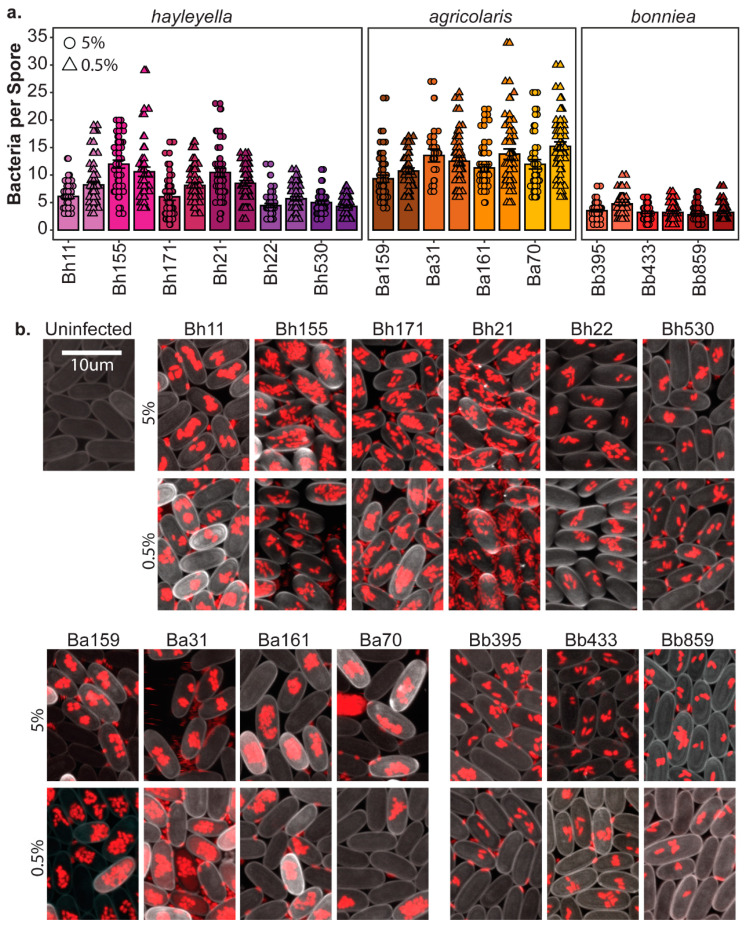
*Paraburkholderia* symbionts result in variable intracellular infection densities. (**a**) The intracellular density of *Paraburkholderia* cells in spores from fruiting bodies developed after exposure to 5% (circles) or 0.5% (triangles) of the indicated *Paraburkholderia* strain as the number of *Paraburkholderia-*RFP cells visualized in individual spores. Points represent individual spores (*n* = 40), bars indicate mean, and error bars are standard error of the mean. (**b**) Representative confocal images of *D. discoideum* spores (cell wall stained with calcofluor pseudo-colored grey) infected with *Burkholderia* cells (fluorescent red) after development on 5% (top panels) or 0.5% (bottom panels) of the indicated strain. Images are Z projections of multiple 5-micron increment slices, scale bar = 10 um.

**Figure 4 genes-11-00674-f004:**
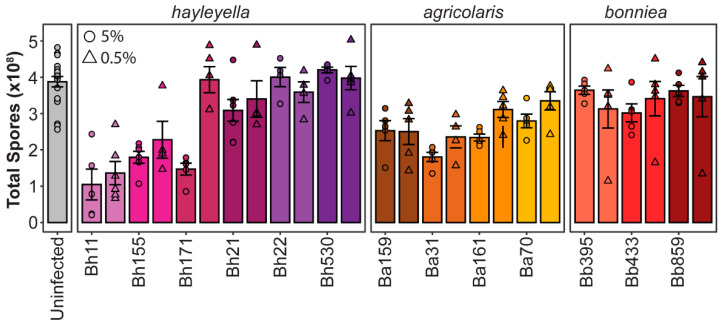
*Paraburkholderia* symbionts differentially alter host spore productivity. Total spores produced after exposure to 5% (circles) or 0.5% (triangles) of the indicated *Paraburkholderia* strain. Points represent individual replicates (*n* ≥ 4), bars indicate mean, and error bars are standard error of the mean.

**Figure 5 genes-11-00674-f005:**
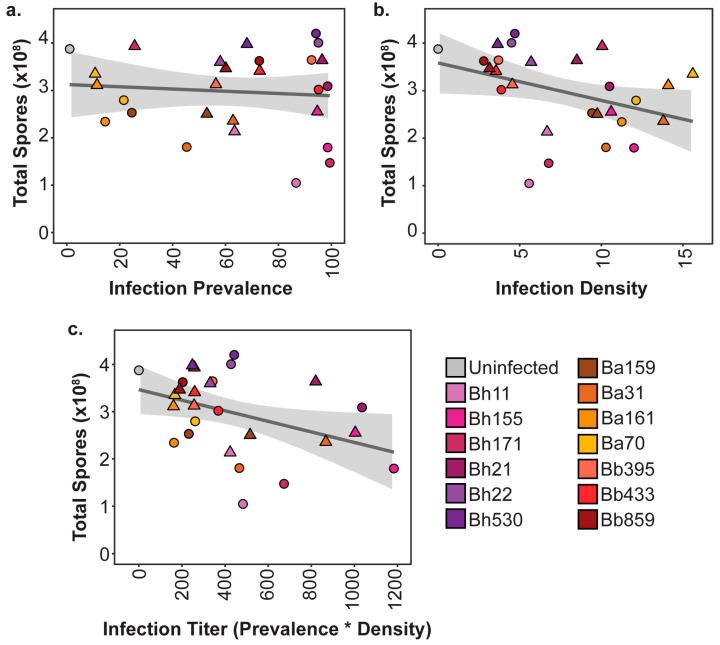
Symbiont infection patterns are not strongly correlated to final host outcomes. (**a**) Correlation between the average total spore productivity and the average infection prevalence; (**b**) average infection density; (**c**) final infection titer (average infection prevalence * average infection density) for 5% (circles) or 0.5% (triangles) symbiont exposure condition.

**Figure 6 genes-11-00674-f006:**
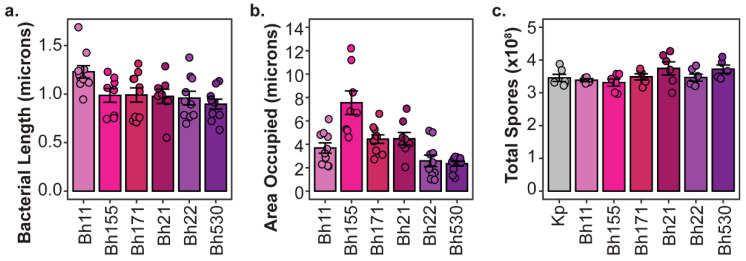
Variation in *P. hayleyella* Characteristics of Potential Consequence to Host Fitness. (**a**) Cell Length of *P. hayleyella* symbiont strains. (**b**) Area occupied by *P. hayleyella* symbionts in individual infected spores from 5% exposure conditions. (**c**) Total spores produced after exposure to filtered supernatants from overnight broth cultures of *P. hayleyella* symbiont strains. For all plots, points represent individual replicates (*n* ≥ 5), bars indicate mean, and error bars are standard error of the mean.

**Figure 7 genes-11-00674-f007:**
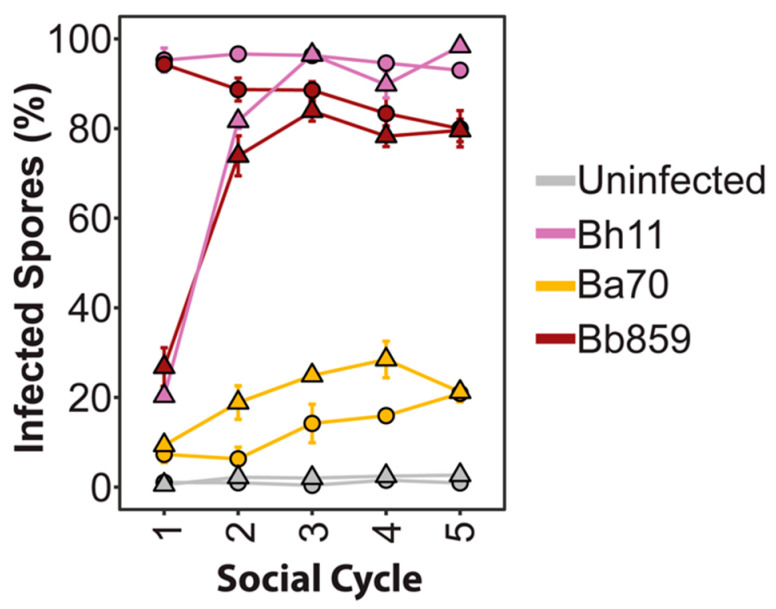
Infection prevalence stabilizes over host developmental cycles. Infection prevalence after multiple social cycles (transfer 1–5) with the first transfer recorded after initial exposure to the indicated *Paraburkholderia* strain at 5% (circles) or 0.5% (triangles) exposure dose. Spores from transfers two through five were subsequently developed on food bacteria alone for transfers. Each data point contains the average of three biological replicates.

**Figure 8 genes-11-00674-f008:**
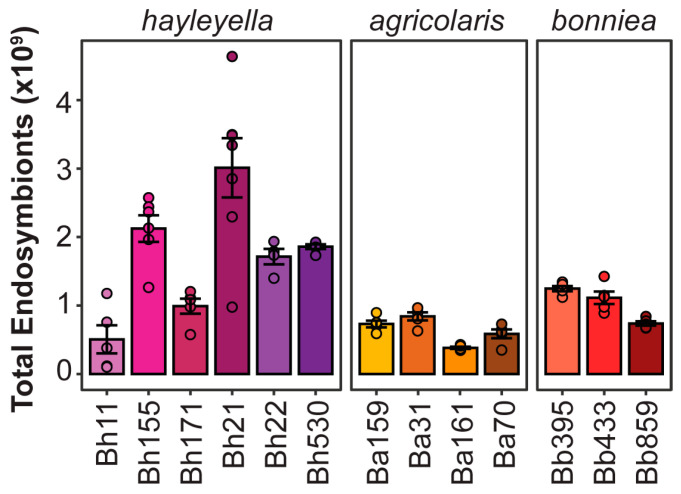
Symbionts with high infectious titers but low fitness costs are most populous. Total intracellular symbiont abundance per host population as estimated by infection titer and host spore productivity. Each total spore measurement was multiplied by the calculated infectious titer (*n* ≥ 4), bars indicate mean and error bars indicate standard error of the mean.

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
