# Peer review of "Paraburkholderia Symbionts Display Variable Infection Patterns That Are Not Predictive of Amoeba Host Outcomes"

_genes, 2020, doi:10.3390/genes11060674_

Round 1
Reviewer 1 Report
The paper is reporting an interesting set of data, related to the bacteria-amoebe interactions. These findings are important as model in microbiome studies.
However, there is a main weakness in the study which limits biological discussion on the findings: which differences have the different strains? Phenotypic analysis on the Parabulkholderia strains must be performed, in the aim to undestand their fitness in non-symbiotic conditions. Moreover, genetic differences (e.g. genome sequences) should be produced, to evaluate if the pattern observed has a coherence with strains'phylogenetic relatedness. A table with strains' features (including isolation etc.) is missing.
Reviewer 2 Report
The study by Miller et al. presents a detailed analysis of 13 strains from three species of the genus Paraburkholderia known for its symbiotic association with the amoeba Dictyostelium discoideum. Their study, which analyzed different aspects of this amoebas-bacteria interaction, did not find clear links between the parameters of the infection and how the host responds to this interaction. However, the approach used by the authors is of interest and the results help to have a better view on the complexity of this symbiotic interaction.
Elements to consider in order to improve the manuscript are described below.
Major comments:
- Lines 95-96: What are the optical density used for each strain? Was this controlled for each experiment? This may be an aspect contributing to the variability of the results observed.
- Lines 138-142: What about the short-term toxicity of the supernatant? Did you have a positive control confirming potential toxicity?
- Lines 159-160: Is it possible that other D. discoideum strains would have a different outcome? It seems pertinent to test, let say, two additional D. discoideum strains with two representative bacterial strains for each bacterial species included in the study.
- Line 173: Discuss more in detail the great variability of the experiments for some bacterial strains.
- Figure 2: Bacterial cell size seems to vary from one strain to another especially for Bh strains. Do this can have an impact on the interaction with amoebas and the infectivity capacity?
- Lines 230-231: Maybe it could be interesting to consider not just the number but also the volume occupied by the symbionts in the spores since not all bacteria have the same size based on figure 2. Larger bacteria potentially represent a higher metabolic load for the host.
- Lines 266-270: The test is not appropriate. Do short-term exposure assay like Cosson et al. 2000 (https://jb.asm.org/content/184/11/3027.long) and include a positive control.
- Lines 307-311: As mentioned earlier, consider the size of the bacteria in your analysis.
- Lines 366-368: You should address this point using a protocol similar to the killing protocol used by the group of Pierre Cosson (https://onlinelibrary.wiley.com/doi/full/10.1111/j.1462-5822.2010.01532.x).
- Lines 381-384: The authors should see the work of Pierre Golstein on this topic in Dictyostelium and they should develop more this aspect.
Minor comments:
- Line 12: You should write: “…influenced, among others, by…”.
- Line 37: replace “single” by “single-cell”.
- Line 38: “In the vegetative stage, single-celled amoebas prey on…”.
- Line 56: “…with P. hayleyella appears to be the…”.
- Lines 78 to 85: Remove the last paragraph of the introduction.
- Lines 92-93: “…we suspended bacteria in stationary phase from SM/5 agar medium…”.
- Line 97: Replace “5/95% and 0.5/99.5%” by “5% and 0.5%”.
- Line 99: “under low light”: What does this mean? No light?
- Throughout the text, put a space between a numerical value and its unit. Also write µ instead of u.
- Line 111: Give URL or reference for the Fiji software.
- Line 210: The sentence is not clear. You should say: "... from one another species."
- Figure 6: Please indicate to which dosage correspond circles and triangles in the graph.
- Lines 295-296: Do we consider that as technical or biological replicates? Please specify.
- Lines 342-343: Is it something that would be worth doing as an experiment in the future based on your results? If yes, indicate it.
- Lines 355-357: This important information should also be mentioned in the M&M section.
- Lines 370-372: The following two paragraphs should be reversed to facilitate reading and better follow the logic of the justification.
- Line 415: The content of the supplementary materials should be explained.
Reviewer 3 Report
The introduction is well written, brief and clear although the authors clearly need to eliminate lines 78 to 85.
Also, in the manuscript they investigate whether secreted toxins could explain differences in host outcomes, but they do not mention in the introduction anything about the importance and the role that toxins may play during infection. I think they should add a brief explanation so the reader can understand better why they decided to investigate further the secreted toxins.
Material and Methods are clearly written and with enough details
Authors should add/discuss more about the genetic background of the different strains used in this study in general, and in particular for those differences that could be relevant to this study.
Authors should comment further on differences observed between the 5 and 0.5% conditions. Some strains behave the same whereas for others there is a big change in infection prevalence. And it does not seem to be related to the number of bacteria/spores. Particularly interesting the drop observed in Bh171, Ba159 and Bb395. Are perhaps these differences related to any particular genes? What are the characteristics of the symbiont genotypes?
On Figure 1, out of the infection prevalence values, one value at 0.5% in Bh171 is really high compared to the other three values. On the other hand, there is a very low value at 0.5% in Bh11, and two values very distant apart, also at 0.5%, in Bb395. Any thoughts for that?
I suggest authors discuss for each strain whether there is a positive/negative correlation between infection prevalence and total spore productivity, as well as, a positive/negative correlation between infection density and spore productivity. As mentioned earlier I wonder to what extend the symbiont genotypes may play a role in these correlations, if they ever happen and are significant. For instance, authors should analyze and discuss why for some strains with high infection prevalence there is a very low total number of spores, whereas for some strains with high infection prevalence there is still a high total number of spores. Another suggestion is to study the outsider strains and see what is different in their genomes compared to the others according to data in Figure 4.
Authors need to provide evidence that the supernatant contains secreted toxins, and that they are active. For instance, by running an assay in parallel to the experiments done in Figure 5 where they can prove both the presence of the toxins and that the effects observed are due to the toxins.
The authors referred several times about the symbiont genotypes but they do not provide any information regarding the similarities or differences between them, not even in the context of infection of Dictyostelium. They need to add it.
Minor comments / Typos:
- Line 89. Replace 0 for O in H2
- Check the spelling of volumes through the entire manuscript. For instance, 200ul (line 98), 500 ul (Line 120), 500uL (Line 139).
- Check the spelling of O.D 600nm (Line 94) compared, for instance, to lines 131 or 139..
- Line 197. I guess author meant pseudo instead of psuedo.
- Line 204. Delete is.
- Line 229. There instead of their.
- Line 238. Replace (b) by (c).
- Figure 6. Indicate 5% (circles) and 0.5% (triangles) in the figure.
Round 2
Reviewer 1 Report
Authors have satisfactorily replied to all my comments
Reviewer 2 Report
I consider that the authors have made significant and satisfying modifications to the manuscript. I consider it now acceptable for publication.
Reviewer 3 Report
The authors incorporated all my suggestions and comments on the manuscript. I have no further comments.